# Early Diagnosis and Classification of Fetal Health Status from a Fetal Cardiotocography Dataset Using Ensemble Learning

**DOI:** 10.3390/diagnostics13152471

**Published:** 2023-07-25

**Authors:** Adem Kuzu, Yunus Santur

**Affiliations:** 1Department of Software Engineering, Firat University, Elazig 23119, Turkey; ademkuzu@gmail.com; 2Department of Artificial Intelligence and Data Engineering, Firat University, Elazig 23119, Turkey

**Keywords:** ensemble learning, fetal health, FHR, NST

## Abstract

(1) Background: According to the World Health Organization (WHO), 6.3 million intrauterine fetal deaths occur every year. The most common method of diagnosing perinatal death and taking early precautions for maternal and fetal health is a nonstress test (NST). Data on the fetal heart rate and uterus contractions from an NST device are interpreted based on a trace printer’s output, allowing for a diagnosis of fetal health to be made by an expert. (2) Methods: in this study, a predictive method based on ensemble learning is proposed for the classification of fetal health (normal, suspicious, pathology) using a cardiotocography dataset of fetal movements and fetal heart rate acceleration from NST tests. (3) Results: the proposed predictor achieved an accuracy level above 99.5% on the test dataset. (4) Conclusions: from the experimental results, it was observed that a fetal health diagnosis can be made during NST using machine learning.

## 1. Introduction

Fetal deaths occurring between the 22nd week of pregnancy and the first 7 days after birth are known as perinatal mortality [1]. The World Health Organization (WHO) has reported that 6.3 million perinatal deaths occur every year. In this report, perinatal cases are scaled to a value per 1000 births, and the development rates of countries are considered to be a critical factor affecting this rate. While this value is lower than 10 in developed countries, it is higher in less developed ones and exceeds 30 in underdeveloped countries [2].

Perinatal cases may result in maternal poisoning/death as well as death for the fetus/infant. The most effective method of early diagnosis is to perform a periodic nonstress test (NST). From the 32nd week of pregnancy, an NST procedure is required every week. The NST device is typically connected to a pregnant woman for 20 min, with two separate probes recording data on fetal heart rate (FHR) and uterine contractions (UCs) [3,4]. The NST device has a memory unit and generates a trace by obtaining FHR and UC data with respect to a basal average of 10 s, as shown in Figure 1.

FHR values range between 30 and 240 on the trace output shown in Figure 1 and can be considered timeseries data. A specialist can take early precautions when necessary after making a diagnosis of the health of the fetus based on this chart. An FHR value of between 120 and 150 is considered normal; however, if it is below 120, this could indicate bradycardia, and if it is above 160, tachycardia may be occurring [5]. The UC value represents the number of contractions within 10 min, and ranges between 0 and 99 [6]. At the same time, acceleration and deceleration occur in the speed of the FHR together with UCs. There are three types of deceleration, which are known as early, late and variable [7]. When UC pressures and these peak changes in FHR are interpreted together from the NST data, it is possible to obtain more information about fetal health. In this case, two (normal, pathological) or three (normal, suspect, pathological) patterns should be considered [8].

Experts evaluate fetal health based on the trace output obtained from the NST device, which has an internal printer, in a hospital environment. By labeling the numerical data from the internal memory of the NST device, a diagnosis can also be made via machine learning. With this in mind, supervised learning approaches can be applied to the task of FHR pattern classification [9]. A supervised learning approach is based on a process of training a model using a machine learning network using labeled data [10]. After the training process, an accuracy and error analysis is performed by creating a confusion matrix for the test and validation data. The use of artificial intelligence approaches in fetal health is not a completely new topic in the literature; studies of the classification of ultrasound images, assistant applications for pregnant women, and pregnancy predictions are common in the fields of image processing and deep learning approaches [11,12,13,14,15,16,17].

In this study, we propose a fetal health classification scheme based on a cardiotocography (CTG) dataset using ensemble learning. The main contributions of our work are as follows:We utilize class weighting as a preprocessing step to avoid overfitting during the model training process.We propose an ensemble learning (EL) model that benefits from the use of ensembles, which can improve the average prediction performance over that of any contributing member in the ensemble.We propose a new multiclass fetal health classification.Our scheme obtains higher accuracy than others in the literature.

## 2. Materials and Methods

In this study, multiclass classification models were developed as binary and multiclass classification networks using linear regression (LR), random forest (RF) and XGBoost algorithms and were applied to the UCI dataset [18]. High accuracy rates were obtained with EL approaches in general [19].

### 2.1. Dataset

The dataset considered in this study contains 2126 fetal CTG records that were automatically processed, and the respective diagnostic features measured. The CTGs were also classified by three expert obstetricians, and a consensus classification label was assigned to each of them. The classification scheme used three fetal states (N: normal; S: suspect; P: pathological). The dataset was therefore used in three-class experiments in this study.

### 2.2. Exploratory Dataset

Multiclass FHR classification was performed using the UCI CTG dataset, which includes FHR and UC data [18]. This dataset consists of 2126 pieces of datum with 22 columns. A screenshot of the sample values from the dataset is shown in Figure 2 in the form of a data frame, extracted with the Pandas library in the scientific Python environment.

The 21 numerical values in the dataset were normalized to the range {from −1 to 1} to create a heatmap. The CTG data attributes and their statistical summaries are shown in Table 1.

CTG is a highly imbalanced dataset, as the total of 2126 samples includes 1655 normal, 295 suspect and 176 pathological entries, as shown in Figure 3.

The correlations between CTG data attributes are shown in Figure 4 in the form of a heatmap. The FHR patterns were coded as normal (0), suspect (1) and pathological (2) in the dataset. In this experimental study, a hybrid network model was developed, and 10-fold cross-validation tests were performed in which we used RF for bagging, XGBoost for boosting and LR for stacking.

### 2.3. Proposed Method

The fetal health outcomes (0: N, 1: S, 2: P) in the dataset were used as the target of the model, and the other data were used as input. A diagram of the overall structure of the pipeline of training and testing of the proposed model is shown in Figure 5.

### 2.4. Data Processing

In the preprocessing step, the numerical input values in the dataset were normalized to the range from −1 to 1 in order to express the correlations in the model’s training. Feature extraction was performed, and adaptive class weighting was used to handle the unbalanced dataset. Equation (1) shows the normalization and standardization process for feature extraction, where $x^{\prime }$ denotes the new value obtained from the *x* values.

The CTG data in Figure 3 show that this is a highly imbalanced dataset. One of the simplest ways to address this class imbalance is to simply provide a weight for each class in order to place more emphasis on the minority classes, such that the result is a classifier that can learn equally from all classes. In a tree-based model, where the optimal split is determined according to some measure such as decreased entropy, we can simply scale the entropy component of each class by the corresponding weight to place more emphasis on the minority classes [19]. The generalized form of the polynomial expansion (PE) function used for feature extraction is given in Equation (2), where *n* represents the degree of expansion and {x,y} represent the independent variables in the dataset. In PE, the dataset grows exponentially horizontally depending on the number *n*, but since the n degree was chosen as 2 in this study, the computational complexity and horizontal growth remained at a minimum level. The entropy function (e) of a class can be calculated as shown in Equation (3), and classes can be dynamically weighted in this way. In imbalanced dataset applications, the oversampling approach is not very practical for high-dimensional data, whereas the undersampling approach removes majority examples to make the majority class the same or a similar size to the minority class, resulting in a significant loss of data. Hybrid approaches require more computation.
(1)$$x^{\prime }=\frac{x-min(x)}{\max\left(x\right)-min(x)}$$
(2)(x+y)n=∑nkxkyn−k
(3)$$e=\sum_{i}p_{i}\log\left(p_{i}\right)$$

### 2.5. Ensemble Learning (EL)

In ML, ensemble methods use multiple learning algorithms to obtain better predictive performance than could be obtained from any of the constituent learning algorithms alone [20].

EL algorithms use two different approaches called bagging and boosting. These techniques help in reducing the variance (bagging—RF) and bias (boosting—XGB) and can improve predictions. Parallel learning takes place in the bagging approach, and sequential learning in the boosting approach. After the learning phase, the errors and weights are updated for the next bootstrap. The stacking process combines predictions from two or more classifiers. RF can use another learning approach that optimally combines stacking learning approaches when voting for the final prediction. Another advantage of EL approaches is that the training and testing times are shorter than those of artificial neural networks [20].

#### 2.5.1. LR

LR is not an EL method but is described here to allow for easier understanding of other methods. The dataset includes *x* (observations) and *y* (dependent variables) {*y_i_*, *x_i_*, *i* = 1,2 … *n*}. The linear relationship between them is shown in Equation (3). The linear relationship between the {*y_i_*, *x_i_*, *i* = 1,2 … *n*} dependent *y* and observations *x* in the dataset is shown in Equation (4).
(4)$${y\prime }_{i}=b+\sum_{i=1}^{n}\omega_{i}x_{i}$$

Here, $y^{\prime }$ represents the estimation and *b* represents the error. It is the difference between the expected value of an estimator and the true value being estimated. The training of the model begins with random or predefined weights *w*. The aim is to reduce the error over the iterations. For this purpose, an error analysis is performed based on the difference between the estimation and the actual value, as given in Equation (5), for the mean square error (MSE). The calculated error value should converge to zero. In this direction, the network weights are updated in the following iteration [21]:(5)$$MSE=\frac{1}{N}\sum_{i=1}^{n}{\left({y_{i}-y\prime }_{i}\right)}^{2}$$

#### 2.5.2. RF

During the training phase, the scheme works on the principle of casting a vote by creating a large number of decision trees. This approach is effective against overfitting, which arises due to the use of a large number of decision trees. After the training process on the selected bootstrap with results (f′) and inputs (x′), the estimation process is calculated based on the average, as given in Equation (6).
(6)$$f\prime =\frac{1}{B}\sum_{b=1}^{B}f_{b}x\prime$$

The updating process shown in Equation (7) is similar to that used in LR, except that in RF, σ is reduced without increasing the bias. Here, *B* is the bootstrap number. Bootstrapping is a statistical resampling technique that involves random sampling of a dataset with replacement [22].
(7)$$\sigma =\sqrt{\frac{{\sum_{b=1}^{B}{(f}_{b}{(x}^{\prime })-f\prime )}^{2}}{B-1}}$$

#### 2.5.3. Gradient Boosting (GB)

Gradient boosting is a machine learning technique used in regression and classification tasks, among others. This is a method of transforming weak learners into strong ones via an incremental process with iterations. The difference between boosting algorithms often lies in how weak learners identify their shortcomings [23].

#### 2.5.4. XGBoost

This improves the performance of the GB algorithm. The main purpose of the regression and supervised-based classification process is to achieve the target function, $f\left(\emptyset \right),$ given in Equation (8).
(8)$$f\left(\emptyset \right)=L\left(\emptyset \right)+\Omega (\emptyset )$$

Here, *L* is the loss function and ∅ is the regularization term. The loss function can be expressed in terms of the MSE, using a similar approach to that of the LR approach, or by using the logistic loss shown in Equation (9) [22].
(9)$$L\left(\emptyset \right)=\sum_{i=1}^{n}\left(y_{i}\ln\left(1+e^{-{y^{\prime }}_{i}}\right)+(1-y_{i})\ln\left(1+e^{{y^{\prime }}_{i}}\right)\right)$$

Similar to the RF approach, the updating process for each selected bootstrap is performed as shown in Equation (10).
(10)$${y\prime }_{i}^{t}=\sum_{k=1}^{t}f_{k}\left(x_{i}\right)={y\prime }_{i}^{t-1}+f_{t}\left(x_{i}\right)$$

### 2.6. Model Evaluation

The evaluation process is carried out using the confusion matrix in Figure 6 and is based on a comparison of true classes with estimations made based on the data that are not used in the training process. The confusion matrix allows for not only the accuracy (Acc) but also the F1-score, precision, recall, and Type-1 (FN) and Type-2 (FP) error conditions to be interpreted [24]. In the Figure 6, true classifications are shown in green, and incorrect classifications are shown in red.

By using the confusion matrix in Figure 6, the performance metrics are obtained as shown in Equations (11)–(14). True positive (*TP*) is the number of instances that were actually positive and were correctly classified as positive. True negative (*TN*) is the number of instances that were actually negative and were correctly classified as negative. False positive (*FP*) is the number of instances that were actually negative but were incorrectly classified as positive. False negative (*FN*) is the number of instances that were actually positive but were incorrectly classified as negative and *N* is the total test sample size. Measuring the performance based on Acc alone is not sufficient, especially for unbalanced datasets. For this reason, values such as recall, precision and $G_{mean}$, which better express the evaluation matrix, are also used. The given equations are for binary classification. However, the formulas for the confusion matrix shown on the right in Figure 6 can be generalized to any number of classes. In this case, the number of estimated and actual values along the diagonal will be equal to the number of classes.
(11)Acc=(TP+TN)/N
(12)Precision=(TP)/(TP+FP)
(13)Recall=(TP)/(TP+FN)
(14)$$G_{mean}=\sqrt{PrecisionxRecall}$$

### 2.7. Weighted Majority Voting

Soft voting and hard voting are two principal distinct methods for combining the predictions of multiple base classifiers in an ensemble model as shown in Figure 7. In addition to these, ensemble models can also compute a weighted majority vote by associating a weight *w_i_* with classifier *M_i_*, as shown in Equation (15), which was used in this study.
(15)$$y={\arg max}_{i}=\sum_{i=1}^{n}w_{i}M_{i}$$

## 3. Results

The results for the evaluation metrics for all of the methods considered here are shown in Table 2. The relative sizes of the training and test datasets were chosen as 0.75 and 0.25, respectively. For the results obtained, the dataset was created randomly each time. The best results were obtained from XGB despite randomly generated datasets and the increase in the test sets.

The evaluation matrices for the experimental results obtained from applying cross-validation to the test dataset are shown in Figure 8, and the results of these algorithms are provided in Table 2.

## 4. Discussion

Studies previously conducted with the same dataset (CTG) related to fetal health classification were considered, and in the following, we examine these in chronological order [18].

In 2021, Fasihi et al. used a 1D convolutional neural network (CNN) and classified fetal health into three patterns with an accuracy of 91%. In their study, they used a new network architecture for the 1D CNN. They examined the performance of this architecture by using five different clinical datasets and observed that the best results were obtained with the 1D CNN [25].

Piri et al. aimed to identify the condition of the fetus and to identify the factors affecting fetal health. They analyzed the original CTG datasets, which were initially unstable, by applying SMOTE. They also found that the performance of the SVM, RF, decision tree (DT) and KNN models improved with a balanced dataset and reported that an improvement of 95% was achieved with RF [26].

Amin et al. used the nearest neighbor, neural network and DT algorithms with the WEKA (Waikato Environment for Knowledge Analysis) application to analyze the CTG dataset and proved that RNN was more efficient than the other machine learning models in terms of classifying the CTG data. They emphasized the superiority of the RNN model, which achieved an accuracy of 95.1% [27].

Kasım applied the extreme learning machine (ELM) algorithm to the CTG data. This author classified the data obtained in this way as normal, suspicious and pathological, as well as well-intentioned and malicious. The performance accuracy of this method was measured based on the F1-score, Cohen’s kappa and recall metrics, and was calculated as 99.29% [28].

Bhowmik et al. employed stacking EL, RF, DT and the extra trees and deep forest classifier algorithms to determine their performance on CTG data. They found that the stacking EL classifier provided an accuracy of approximately 96.05% in estimating fetal health risks [29].

Haweel et al. examined the growth and safety conditions of the fetus with electronic fetal monitoring (EFM). They used the Newton least mean squares (NLMS) algorithm for training of a polynomial neural network (PNN). At the end of the experiment, they found a classification accuracy of 99.74%. They compared the performance of the PNN classifier with functional link artificial neural network (FLANN) classifiers such as a Legendre neural network (LNN) and a Volterra neural network (VNN) and found that the PNN classifier provided better performance [30].

In 2020, Fei et al. proposed FCM-ANFIS (fuzzy C-means clustering based adaptive neuro-fuzzy inference system) to automatically classify CTG data. The rationale for this was that they found that the FCM (fuzzy C-means clustering-based) algorithm had not previously been used for the fuzzy parts of existing ANFIS models. Based on their experimental results, they determined that the FCM-ANFIS model could remove uncertainty and complexity from CTG interpretations and found that it could learn and adapt [31].

Nandipati et al. aimed to evaluate classification models and feature selection on CTG using CARET and Scikit Learn. They observed the best results with RF but found that naive Bayes, with its reduced attributions (Python-ML techniques), achieved an accuracy rate of between 91.88% and 100% [32].

John et al. examined the condition of the fetus with naive Bayes, RF, J48 and stacking models using CTG data. With the stacking model, they achieved a more balanced accuracy than with the other algorithms and observed that this model most accurately predicted the pathological fetal condition, with an accuracy of 98.9% [33].

Piri et al. aimed to identify the most significant causes of fetal death with the evolutionary multi-objective genetic algorithm (MOGA). They proposed an approach called MOGA-CD to obtain the best information about the health status of the fetus. They observed that DT, RF, Gaussian naive Bayes (GNB), SVM and extreme gradient boosting (XGBoost) achieved the best performance in terms of the classification accuracy of the fetal health status on reduced datasets [34].

In 2019, Chen et al. aimed to monitor the growth of fetuses by creating an intelligent model with unstable CTG data. They adjusted the category weights of the weighted RF (WRF) model and tested the resulting model on the CTG dataset retrieved from the UCI repository. They compared their test results with those of 10 other discriminant fetal monitoring models. The WRF model achieved a good performance of 97.85% based on the F1 score. These authors also proved the superiority of the WRF model for the classification of unbalanced VTG data [35].

Sevani et al. aimed to evaluate the level of correlation between the CTG data and their characteristics. They used SVM as a classifier and assessed its performance based on the F1 score method. They also measured the performance on other datasets and determined that this method could be applied successfully. As a result of their experiments, they observed that the accuracy of the fetal state increased from 94.35% to 99.91% [36].

Katuwal et al. were inspired by the better performance of random vector functional link (RVFL) on an ELM. They proposed several deep RVFL variants using a framework of stacked autoencoders, in which direct connections were created from the previous layers of the original RVFL network to the front layers of the networks. They observed that these connections made the randomization more regular and that this reduced the complexity of the model [37].

Vani used deep learning neural networks and the decision support system to determine the health status of fetuses. He applied metrics such as specificity, F1 score and G-mean to measure the performance and accuracy. He observed that the SVM model outperformed the others with an F1 score of 93% and an accuracy of 81%. He also reported that the DNN model had higher performance than SVM, with a G-mean of 91% and a sensitivity of 89% [38].

Mohammad Saber Iraji proposed artificial intelligence-based approaches to predict the condition of fetuses. NN, DSSAEs, Deep-ANFIS and MLA-ANFIS topologies were applied to the dataset, and the best estimate of fetal health status was obtained from the DSSA method, with an accuracy of 99.503% [39].

In 2018, Uzun et al. aimed to classify the CTG dataset, which consists of 21 features and 2126 different signals, using the ELM method. The classification criteria were based on A-SUSP (a morphology model). The results of their experiments indicated that the highest accuracy of 84.3% was obtained from the PCA-14 ELM machine learning algorithm [40].

Deressa et al. aimed to select the most efficient data mining algorithm to predict the health of the fetus from the CTG dataset. They used artificial neural network methods based on RF, C4.5 (J48), SVM and genetic algorithms. Their results showed that hybrids of two or more algorithms improved the performance of classifiers in predicting the treatment of fetal movements [41].

In 2022, Mehbodniya et al. used ML algorithms to predict the health state of the fetus, including random forest (RF), K-nearest neighbors (KNN), support vector machine (SVM) and MLP. From the classification results, it was evident that GB outperformed all the other classifiers in this study [42].

In 2023, Kaliappan et al. used ensemble voting classifier algorithms for early detection of the health state of fetuses, including Adaboost, KNN, Gaussian naïve Bayes (GNB) and Monte Carlo cross-validation. From the classification results, it was evident that RF outperformed all the other classifiers in the study [43].

In 2023, Shrutki et al. used ensemble voting classifier algorithms, including random forest (RF) and the genetic algorithm (GA), to classify fetus health. From the classification results, it was evident that RF outperformed all the other classifiers in the study [44].

The results obtained from our study are compared in Table 3 with those of previous studies in the literature for the same dataset. The proposed approach achieved similar or better results compared with the other research. It can be planned to repeat the study in the future using other ML algorithms as another research study [45].

In this study, an EL-based approach using CTG was proposed to diagnose fetal health status. When the proposed approach is compared with other studies from the literature, higher accuracy was achieved. In addition to the proposed approach obtaining high accuracy, it also requires less computational complexity. The limitations of this research can be summarized as follows: First, the dataset used is highly unbalanced. It is impossible to collect new balanced data for research or very difficult to obtain access to the data. Second, there are other datasets where the proposed approach can be used. However, these datasets are not accessible because they are not publicly available. However, within the scope of the project mentioned in the acknowledgment section, designing a wearable and remotely monitorable NST device, collecting new data or designing new ML algorithms could be potential future works in the area.

## 5. Conclusions

In this study, a new approach was presented for pattern classification of fetal health in which EL algorithms were applied to three-class labeled data obtained from NST. Compared to previous studies in the literature, better results were obtained even in cases of cross-validation and class weighting of the dataset. The main advantage of our approach was that the training and testing times were shorter.

To prove that the 100% accuracy value obtained with the XGB algorithm was not simply due to chance or excessive compliance, the dataset was randomly mixed and cross-verified. However, this dataset, consisting of 2126 items, was still unbalanced. Of the data, 1566 were labeled as normal, 295 as suspicious and 176 as pathological. Averaging, filtering and downsampling approaches have been used in the literature to handle unbalanced datasets. In this part of the experimental study, 100 random samples (with relative sizes of 0.75 for the training set and 0.25 for the test set) were taken from each group and re-trained with XGBoost. The results showed an accuracy level above 99%. However, this value dropped dramatically when the scenario was sampled down based on the pathological class, which was the smallest, rather than downsampling all three classes in equal proportion. In the last scenario, although the pathological data were still classified with high accuracy, the classification accuracy of normal and suspicious labeled data changed dramatically. We conclude that more studies should be conducted on this unbalanced dataset in future.

However, a point to consider was that the dataset used in this particular study was unbalanced. With this in mind, in the second scenario, the dataset was downsampled by 95%. To carry out undersampling, the proportions of the data in the three classes were chosen equally, and an accuracy above 100% was still achieved. However, the overall accuracy rate dropped dramatically when the class of pathological data, which had a smaller sample size, was taken as a reference for downsampling. Due to the importance of these results, future studies should focus on classifying unbalanced data by producing synthetic data with generative adversarial networks and statistical approaches, and the results should be compared using oversampling.

## Figures and Tables

**Figure 1 diagnostics-13-02471-f001:**
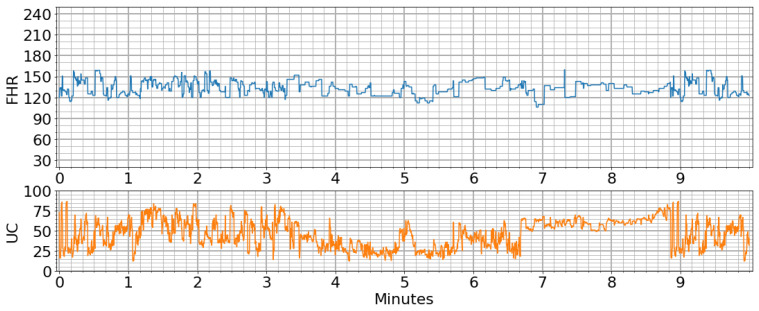
Example of an NST trace.

**Figure 2 diagnostics-13-02471-f002:**
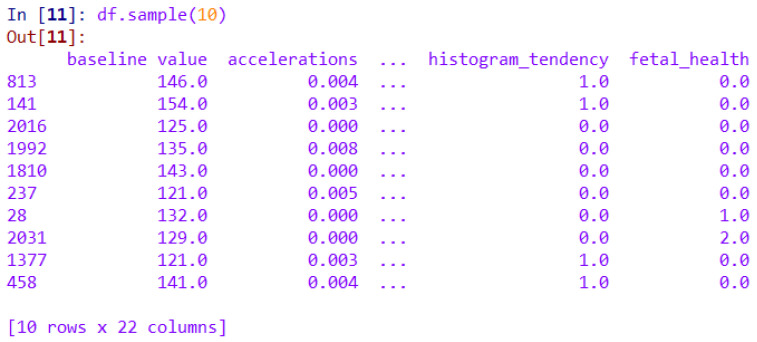
CTG dataset in the form of a Python Pandas data frame.

**Figure 3 diagnostics-13-02471-f003:**
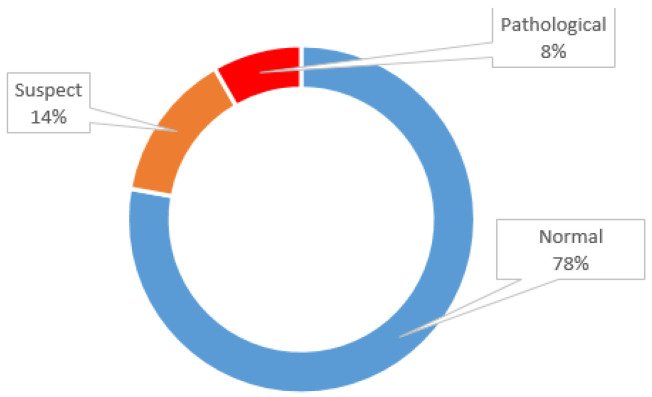
Distribution of the CTG data.

**Figure 4 diagnostics-13-02471-f004:**
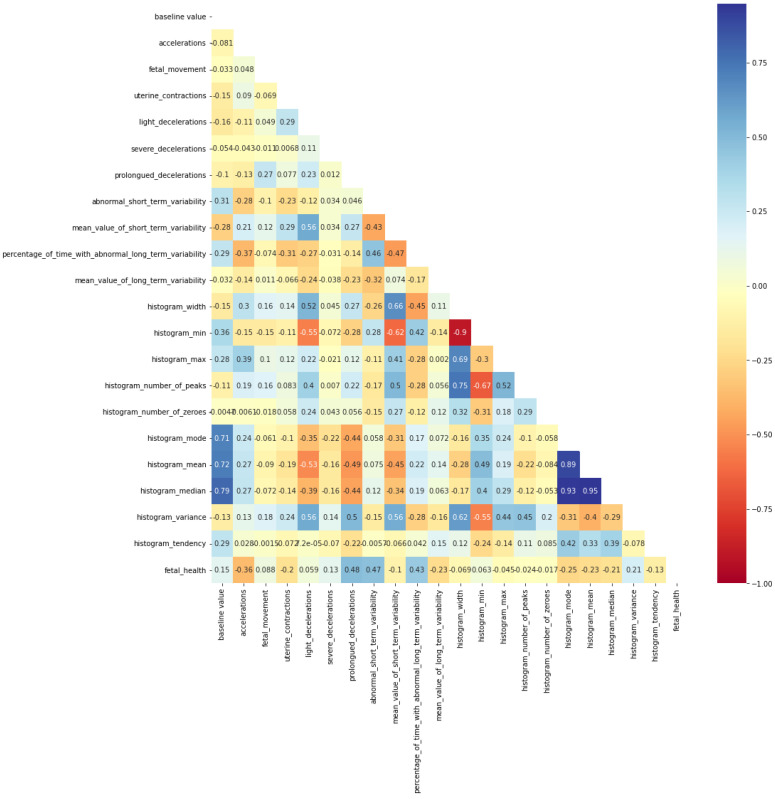
Visualization of CTG correlations as a heatmap: {+1: strong correlation, 0: no correlation, −1: strong inverse correlation}.

**Figure 5 diagnostics-13-02471-f005:**
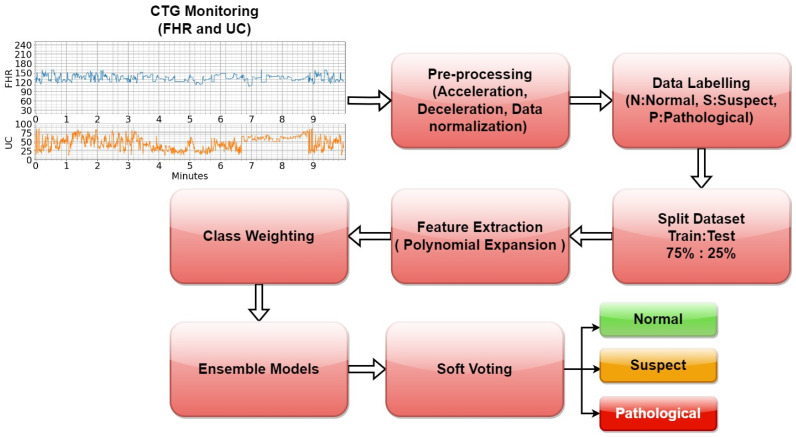
Workflow of the fetal health classification model using ensemble model.

**Figure 6 diagnostics-13-02471-f006:**
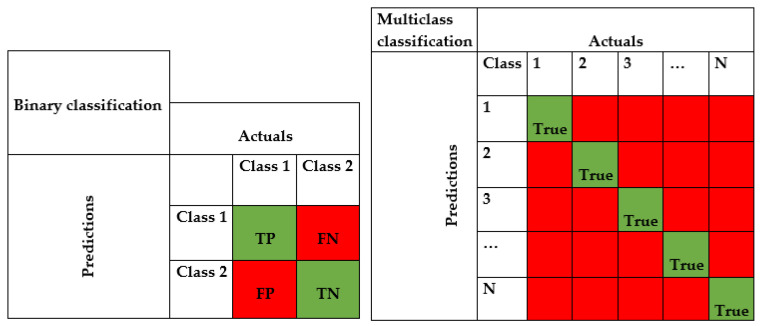
Evaluation matrix for binary and multiclass classification.

**Figure 7 diagnostics-13-02471-f007:**
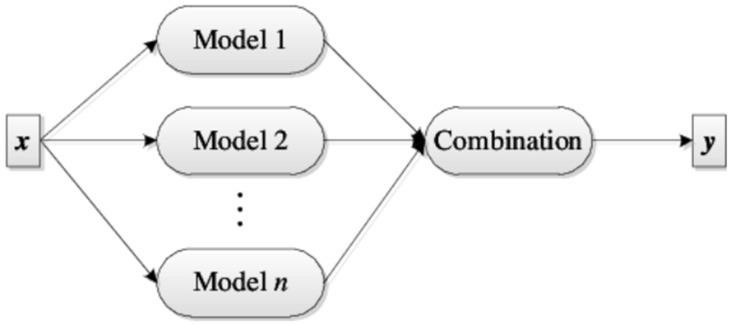
Combining of ensemble models.

**Figure 8 diagnostics-13-02471-f008:**
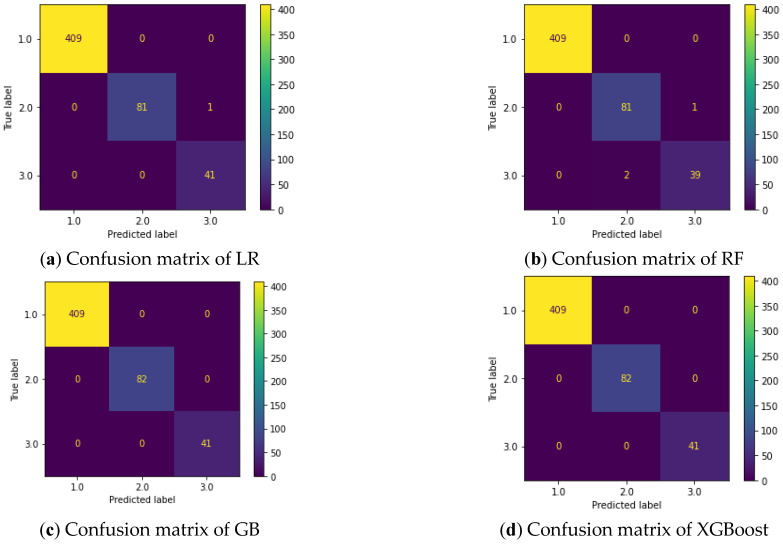
Confusion matrices for the considered methods.

**Table 1 diagnostics-13-02471-t001:** Attributes of the CTG.

Attribute	Description and Unit	Mean	Std	Min	Max
Baseline value	Beats per minute	133.3039	9.840844	106	160
Accelerations	Accelerations per second	0.003178	0.003866	0	0.019
Fetal movement	Fetal movements per second	0.009481	0.046666	0	0.481
Uterine contractions	Uterine contractions per second	0.004366	0.002946	0	0.015
Light decelerations	Light decelerations per second	0.001889	0.00296	0	0.015
Severe decelerations	Severe decelerations per second	3.29 × 10^−6^	5.73 × 10^−5^	0	0.001
Prolonged decelerations	Prolonged decelerations per second	0.000159	0.00059	0	0.005
Abnormal short-term variability	percentage of time with abnormal short term variability	46.99012	17.19281	12	87
Mean value of short-term variability	Mean value of short term variability	1.332785	0.883241	0.2	7
% abnormal long-term variability	Percentage of time with abnormal long term variability	9.84666	18.39688	0	91
Mean value of long-term variability	Mean value of long term variability	8.187629	5.628247	0	50.7
Histogram width	Width of FHR histogram	70.44591	38.95569	3	180
Histogram min	Minimum of FHR histogram	93.57949	29.56021	50	159
Histogram max	Maximum of FHR histogram	164.0254	17.94418	122	238
Histogram number of peaks	Histogram peaks	4.068203	2.949386	0	18
Histogram number of zeroes	Histogram zeros	0.323612	0.706059	0	10
Histogram mode	Histogram mode	137.452	16.38129	60	187
Histogram mean	Histogram mean	134.6105	15.5936	73	182
Histogram median	Histogram median	138.0903	14.46659	77	186
Histogram variance	Histogram variance	18.80809	28.97764	0	269
Histogram tendency	Histogram tendency	0.32032	0.610829	−1	1
Fetal health	Fetal state class (0: normal (N); 1: suspect (S); 2: pathological (P))	-	-	0	2

**Table 2 diagnostics-13-02471-t002:** Performance evaluation of compared methods.

Attribute	Acc	F1	Recall	Precision	G-Mean
LR	0.99	0.99	1	0.99	0.99
RF	0.99	0.99	0.99	0.99	0.99
GB	1	1	1	1	1
XGBoost	1	1	1	1	1

**Table 3 diagnostics-13-02471-t003:** Performance evaluation of the compared literature works using CTG.

Author and Reference Work	Year	Methods	Classification	Accuracy (%)
Fasihi et al. [25]	2021	GA, Rprop, PNN, EL, LS-SVM, SF, XGBoost,	Multiclass	97.46
Piri et al. [26]	2021	SVM, RF, DT, KNN	Multiclass	95.00
Amin et al. [27]	2021	IN-RNNs, RNNs, NNs, nearest neighbor, DT	Multiclass	95.10
Kasım [28]	2021	ELM	Multiclass	99.29
Bhowmik et al. [29]	2021	DT, RF, ET, DF, EL	Multiclass	96.05
Haweel et al. [30]	2021	PNN, KNN, SVM, DT, RF, NB, B&B Model, ANN, LNN	Multiclass	99.30
Fei at al. [31]	2020	FCM	Multiclass	96.39
Nandipati et al. [32]	2020	KNN, SVM, RF, NB, NN, B&B, feature selection approaches	Multiclass	95.07
John et al. [33]	2020	NB, RF, J48 (C4.5), stacking model	Binary class	98.90
Piri et al. [34]	2020	LR, SVM, KNN, XGBoost, DT, RF, GNB	Multi Class	94.00
Chen et al. [35]	2019	WRF, DT, RF, BP, SVM, KNN, opportunity	Multiclass	99.76
Sevani et al. [36]	2019	SVM	Binary class	94.35
Katuwal et al. [37]	2019	RVFL, ELM, AE	Multiclass	99.32
Vani [38]	2019	SVM, RNN, NN, DT, KNN	Multiclass	94.00
Iraji [39]	2019	NN, DSSAEs, deep-ANFIS	Multiclass	99.50
Uzun et al. [40]	2018	ELM	Multiclass	99.18
Deressa et al. [41]	2018	RF, GA, OBFA	Multiclass	93.61
Mehbodniya et al. [42]	2022	SVM, RF, MLP, KNN	Multiclass	94.5
Kaliappan et al. [43]	2023	GB	Multiclass	99.
Shrutki et al. [44]	2023	RF, GA	Multiclass	96.62
Our method		RF, GB, XGBoost	Multiclass	>99.5

## Data Availability

Dataset is provided as [18].

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
