# Peer review of "Early Diagnosis and Classification of Fetal Health Status from a Fetal Cardiotocography Dataset Using Ensemble Learning"

_diagnostics, 2023, doi:10.3390/diagnostics13152471_

Round 1
Reviewer 1 Report (Previous Reviewer 1)
The paper is improved significantly as compared to earlier version.
Author Response
The respected reviewer's comments have guided and contributed significantly to the improvement of our paper in its previous versions. We are sincerely grateful to the reviewer for accepting our article for publication in this prestigious journal.
Reviewer 2 Report (Previous Reviewer 2)
I accept the corrections and wish the authors good luck in their further research!
Author Response
The respected reviewer's comments have guided and contributed significantly to the improvement of our paper in its previous versions. We are sincerely grateful to the reviewer for accepting our article for publication in this prestigious journal.
Reviewer 3 Report (Previous Reviewer 3)
In diagnostics-2478984.R1, Santur and Kuzu attempted to use ensemble learning (EL) to diagnose fetal health status early and to classify it from a fetal cardiotocography (CTG) dataset.
S1. The authors addressed Reviewer 3's previous concerns.
W1. In terms of novelity, the authors simply applied ensemble learning on thee existing classifiers: random forest (RF), gradient boosting (GB) and XGBoost algorithms.
W2. The results also showed that RF was worst than existing non-EL model of linear regression (LR).
W3. Given GB and XGBoost led to the same performance as shown in Table 2, is XGBoost needed?
Author Response
The respected reviewer's comments have guided and contributed significantly to the improvement of our paper in its previous versions. We are sincerely grateful to the respected reviewer. Our responses attached as a file.

This manuscript is a resubmission of an earlier submission. The following is a list of the peer review reports and author responses from that submission.
Round 1
Reviewer 1 Report
The study is interesting but the manuscript needs some more details. Below I have mentioned some specific comments
1. Page 3, Table 1, Units are needed for Attributes.
2. Page 3, Table 1, Commas are used in the place of dots.
3. Page 5, Index are wrong in whole paper (2.2 is mentioned for Exploratory Dataset and preprocessing).
4. Page 5, Figures 5: what feature extraction technique is used? Explain.
5. Page 5, Figures 5: Check spelling for training.
6. Page 5, Figures 5: what is the training testing ratio?
7. Page 7, Index are wrong.
8. Explain about the parameters and hyperparameters of RF and XGBoost
9. Explain about the fine tuning of hyperparameter used for analysis.
10. Page 11, Table 3, Commas are used in the place of dots.
Reviewer 2 Report
The authors did not correct the remarks very painstakingly, one can agree with some of the edits (if not to find fault). In particular, the title of Figure 5 has not been corrected, Table 3 is also almost unedited (only 1 source has been added, but the data sets on which each result was obtained are still not indicated and brief indicators of each data set and computational experiment are not given (number of examples, classes, the size of the test sample, so that you can have an idea of the reliability) I still do not understand why the descriptions of previous works were moved to section 4 (the review part, which contains a lot of text, but actually repeats the information from the table).
I think that these points still need to be corrected.
Reviewer 3 Report
In diagnostics-2478984, Santur and Kuzu attempted to use ensemble learning (EL) to diagnose fetal health status early and to classify it from a fetal cardiotocography (CTG) dataset.
W1. The authors simply applied ensemble learning on thee existing classifiers: random forest (RF), support vector machines (SVM) and XGBoost algorithms.
W2. There is no clear description on the ensemble part.
W3. It is also unclear which classifier in the ensemble was the clear contributor.
W4. The authors only applied to a single UCI dataset [18] consisting of 2,126 fetal CTG records and 23 features. What would be the expected results for other similar dataset?
W5. In Section 2.4, the authors described an evaluation matrix for binary classification. However, the confusion matrices in Figure 7 seem to be for multi-class classification.
W6. What are the three classes? Normal, suspect, and pathological?
W7. Variables in many equations were mainly undefined. For instance, what is N is Eq. (10)?
W8. Performance shown in Table 2 appears to be unrealistically high, with 0.99 or 1 in all entries.
W9. The results also showed that RF was worst than existing non-EL model of linear regression (LR).